# *Toxoplasma gondii* and *Alternaria* sp.: An Original Association in an Immunosuppressed Dog with Persistent Skin Lesions

**DOI:** 10.3390/pathogens12010114

**Published:** 2023-01-10

**Authors:** Radu Blaga, Virginie Fabres, Vincent Leynaud, Jean-Jacques Fontaine, Edouard Reyes-Gomez, Amaury Briand, Odile Crosaz, Isabelle Lagrange, Amandine Blaizot, Delphine Le Roux, Veronica Risco Castillo, Pavlo Maksimov, Jacques Guillot, Jens Peter Teifke, Gereon Schares

**Affiliations:** 1Anses, INRAE, Ecole Nationale Vétérinaire d’Alfort, Laboratoire de Santé Animale, BIPAR, F-94700 Maisons-Alfort, France; 2Ecole Nationale Vétérinaire d’Alfort, CHUVA, Unité de Médecine, F-94700 Maisons-Alfort, France; 3Ecole Nationale Vétérinaire d’Alfort, BioPôle, Laboratoire d’anatomo-cytopathologie, F-94700 Maisons-Alfort, France; 4Ecole Nationale Vétérinaire d’Alfort, CHUVA, Unité de Dermatologie, F-94700 Maisons-Alfort, France; 5Ecole Nationale Vétérinaire d’Alfort, Biopôle, Laboratoire de Biochimie-hématologie, F-94700 Maisons-Alfort, France; 6Institute of Epidemiology, Friedrich-Loeffler-Institut, Federal Research Institute for Animal Health, Südufer 10, 17493 Greifswald, Germany; 7Oniris, Department of Dermatology, Parasitology, Mycology, F-44300 Nantes, France; 8Department of Experimental Animal Facilities and Biorisk Management, Friedrich-Loeffler-Institut, Federal Research Institute for Animal Health, 17493 Greifswald, Germany

**Keywords:** *Toxoplasma gondii*, *Alternaria*, immunosuppressive therapy, skin lesions

## Abstract

Dogs and cats may suffer from a variety of diseases, mainly immune mediated, that require the administration of immunosuppressive drugs. Such therapies can cause adverse effects either by the toxicity of the drugs or as a consequence of immune suppression and associated opportunistic infections. Here we present an, yet unknown, association of *Toxoplasma gondii* and *Alternaria* fungus, within cutaneous lesions in a dog under long-term immunosuppressive therapy. The diagnosis of such infections is laborious and not obvious at first glance, since the clinical signs of cutaneous toxoplasmosis, neosporosis or alternariosis are not specific. A further laboratory confirmation is needed. Therefore, we currently recommend that dogs and cats should undergo serologic testing for toxoplasmosis or neosporosis prior to immunosuppressive therapy and a regular dermatological evaluation during the immunosuppressive therapy.

## 1. Introduction

Dogs and cats may suffer from a variety of diseases, mainly immune-mediated, that require the administration of immunosuppressive drugs. Such a therapy can cause adverse effects either by the toxicity of the drugs or as a consequence of immune suppression and associated opportunistic infections [1]. The clinical signs, the severity and the frequency of the opportunistic infections tend to be related to the dose and the duration of immunosuppressive treatment, making them unpredictable in appearance, and thus difficult to diagnose at first glance.

Here we present an as yet unknown association of *Toxoplasma gondii* and *Alternaria* fungus, within cutaneous lesions in a dog under long-term immunosuppressive therapy. 

*Toxoplasma gondii* is a highly zoonotic intra-cellular protozoan parasite, with a complex life cycle [2]. Domestic cats and other felids act as definitive hosts, able to shed millions of oocysts into the environment. Virtually all warm-blooded mammals and birds are known as intermediate hosts of the parasite. *Toxoplasma gondii* is capable of invading a large variety of tissues and penetrating any nucleus-bearing cell of warm-blooded animals. Acute toxoplasmosis is caused by the rapid-multiplying stage of *T. gondii*, the tachyzoite. In the chronic phase of infection, the parasite is capable of surviving in tissue cysts, enclosing hundreds of bradyzoites, most likely for the life time of its host. Stage conversion, i.e., a conversion of bradyzoites into tachyzoites, in other words a shift from chronic toxoplasmosis into acute toxoplasmosis, is noticed frequently in immunosuppressed patients [3,4,5]. A very few cases of cutaneous infection due to *T. gondii* have been described in dogs [6] or humans [7].

*Alternaria* spp. are filamentous fungi commonly found in soil or associated with plants. In humans and animals, *Alternaria* spp. may be responsible for subcutaneous infection following accidental inoculation through the skin [8,9,10,11]. The definitive diagnosis of *Alternaria* spp. infection requires demonstration of yeast-like elements and/or filaments in histopathological preparations, isolation on fungal culture and PCR analysis [12].

Although several papers reported *T. gondii* or *Alternaria* spp. skin lesions, this paper aims to describe a first case of a concomitant cutaneous toxoplasmosis and *Alternaria* spp. infection in a dog under long-term immunosuppressive therapy, with a special emphasis on diagnosis. 

## 2. Case Report

A 2.5-year-old female Dachshund was presented for persistent skin lesions to the Alfort University Veterinary Hospital, France, in February 2017. The dog had been diagnosed with an immune-mediated hemolytic anemia 1.5 years prior and had been under immunosuppressive treatment (prednisolone 0.9 mg/kg/day and cyclosporine 5.6 mg/kg BID) since then with regular follow-up visits. The clinical examination was unremarkable, showing an afebrile patient with cardiac frequency of 120 bpm, capillary refill time of 2 s and a stable weight (8.6 kg). A systolic murmur identified 6 months prior was persisting at the same intensity: grade 4/6. A history of chronic diarrhea was mentioned but no clinical signs were observed on that day. The dermatological examination revealed the presence of two 1 cm diameter non-pruritic, suppurative nodules on the thorax and on the bridge of the nose as well as several round alopecic nodules on the trunk. Hair loss had been noticed, according to the owner, since the beginning of the immunosuppressive treatment, while the nodules were reported for the first time 2 months ago. Fine-needle aspiration cytology of the lesions revealed numerous crescent-shape elements (5 × 3 μm) consistent with protozoan tachyzoites (resembling *T. gondii* or *Neospora caninum*), as well as yeast-like elements (Figure 1). 

Additional lab findings showed a normal white blood cell (WBC) count (10,360 cells/µL; reference interval (RI), 2900–13,600), 80% of neutrophils, with low eosinophils (80 cells/µL; RI, 100–1500 cells/µL) and low lymphocytes (520 cells/µL; RI 1100–5300 cells/µL) reflecting corticosteroid treatment. A mild regenerative anemia was diagnosed based on hemoglobin concentration (11.6 mg/dL; RI, 12.4 to 19.2 mg/dL), and reticulocyte count (211,500 cells/µL; RI, 19,400–150,100) consistent with persistence of the hemolytic anemia or blood loss. No eggs, oocysts or cysts were identified in the feces by saccharose, MgSO_4_ flotation or biphasic sedimentation. Blood serum tested positive for *T. gondii* antibodies (1:51.200) and negative for *N. caninum* antibodies (<1:50) by in-house Indirect Fluorescent Antibody Test (IFAT).

A punch biopsy of the thorax nodule was performed. Histological examination revealed a pyogranulomatous dermatitis with numerous clusters of protozoan stages, resembling multiplying tachyzoites, and PAS-positive yeast-like elements and mycelian filaments (Figure 2). Indirect immunohistochemistry test of the paraffin-embedded sections incubated with specific anti-*T. gondii* antibodies returned positive. 

Following an overnight digestion of 1/3rd of the punch biopsy with a trypsin medium, one part of the supernatant was used for DNA extraction and the other part was cell cultured on Human Foreskin Fibroblast (HFF) cell line, showing a rapid multiplication of protozoan tachyzoites. They both returned positive results when a *T. gondii* 529 bp RE-target real-time Polymerase Chain Reaction (PCR) [13,14,15] was performed. Punch biopsy DNA revealed a low Ct value of 16.3 in the *T. gondii* real-time PCR which is indicative of a high parasite DNA concentration and is in accordance with the histological observations (Figure 2D). DNA was also examined for the presence of *N. caninum* DNA by a Nc5 gene-target real-time PCR [14,16], but with a negative result. The fungal culture from the punch biopsy yielded negative results, whereas the U1/2-fungal PCR [17] on the DNA extraction of the biopsy gave a positive result. Following cloning and sequencing of the amplification product, *Alternaria* sp. was identified with 100% of identity for this sequence. 

Genotyping by microsatellites and PCR-RFLP of the *T. gondii* tachyzoites issued from the HFF cell culture, and used a previously described method [13] based on 8 markers, which are distributed over eight chromosomes, and the apicoplast [18], namely nSAG2, SAG3, BTUB, GRA6, C22-8, C29-2, L358, PK1, Apico. Comparison with *T. gondii* reference strain DNAs of RH (Type I), Me49 (Type II) and NED (Type III) identified the present strain as Type II, Apico I, resembling ToxoDB#3. 

The dog was treated with clindamycin 11 mg/kg BID for 30 days. In the following three months, the nodules and the alopecia lesions disappeared and the immunosuppressive treatment was reduced (0.12 mg/kg/day of prednisolone, cyclosporine 5 mg/kg BID). During the next 2 years of the follow-up, the systolic murmur improved to a grade of 2/6, no more history of chronic diarrhea was mentioned and the immunosuppressive treatment was stopped. During this time, the *T. gondii* antibody level detection by Modified Agglutination Test (MAT) remained high (1:12.288). 

## 3. Discussion

*Toxoplasma gondii*, a worldwide ubiquitous protozoan parasite, has a complex life cycle with domestic cats and other felids as definitive hosts and virtually all warm-blooded animals as intermediate hosts [2]. *T. gondii* has three infectious stages: oocysts, bradyzoites and tachyzoites. The main route of transmission is the ingestion of tissue cysts present in intermediate hosts, containing bradyzoites. A second route is the oral uptake of sporulated oocysts shed by definitive hosts and contaminated vegetables, soil and water. Third, rapidly multiplying tachyzoites are at the origin of congenital transmission.

Dogs are frequently infected with *T. gondii* but a relatively limited number of clinical cases have been reported [2]. Clinical toxoplasmosis seems to be much more frequent in cats whereas dogs tend to be more often affected by *N. caninum*, a very closely related Apicomplexan parasite [6]. Cases of toxoplasmosis in dogs are related either to canine distemper virus (CDV) infection, which lowers the resistance to a pre-existing *T. gondii* infection or to immunosuppressive chemotherapy [2], like in the present case. Usually, there is a low rate of morbidity and mortality, with pulmonary and nervous forms that are the most common clinical signs of toxoplasmosis in dogs. Some atypical forms with digestive, ocular and muscular signs may occur also [2]. 

Only five cases of cutaneous toxoplasmosis have been reported in dogs worldwide [6] while thirteen cases of cutaneous neosporosis have been identified [19]. Only in one of these cases of cutaneous toxoplasmosis—a Brazilian case—*T. gondii* was genotyped and revealed as Type BrI (ToxoDB#6) using PCR-RFLP. In the present case from France, *T. gondii* Type II (ToxoDB#3) was observed. This is not a surprise because *T. gondii* Type II (ToxoDB#1 and ToxoDB#3) is the prevailing genotype in Europe [20]. To the best of our knowledge, there is no indication that canine hosts have an increased susceptibility to particular *T. gondii* genotypes. 

Immunosuppression was systematically associated with toxoplasmosis cases and this was true only in half of the neosporosis cases. In humans, cutaneous toxoplasmosis has been described in immunosuppressed or transplanted individuals [7], but no case of human neosporosis has been identified so far. 

In the present case, the microscopic examination of the fine-needle aspirate revealed evidence of a protozoan infection, compatible with both cutaneous toxoplasmosis or neosporosis. Clinical signs of cutaneous toxoplasmosis or neosporosis are not specific, and thus need further laboratory confirmation. Currently, routine diagnosis of toxoplasmosis or neosporosis relies mainly on the use of serological assays such as the Sabine–Feldman dye test (only toxoplasmosis), IFAT, enzyme-linked immunosorbent assay (ELISA) or various agglutination tests (like MAT used here). The latter have the advantage of being applicable to serum from any animal species, and not requiring species-specific conjugates, as, e.g., ELISA tests do. Most clinical laboratories use ELISA tests for the routine screening of specific immunoglobulin IgG and IgM, while the use of other serological techniques is limited to reference laboratories [21]. Because in our case the dog tested strongly positive for *T. gondii* antibodies but negative for *N. caninum* antibodies, the presence of a cutaneous neosporosis was unlikely. This was confirmed by a negative real-time PCR from punch biopsy DNA targeting *N. caninum*. However, since *T. gondii* infections are widespread in dogs [2], antibody detection could represent only a coincidence. Kinetics and repeated serological testing at 15 day intervals may allow to suspect a progressive toxoplasmosis. However, in cases of chronic toxoplasmosis, including cutaneous toxoplasmosis, a significant increase of antibody levels within a 15 day interval is unlikely [22]. 

Direct tests like isolation of the parasite, immunohistochemistry or PCR need to be applied to confirm the suspected toxoplasmosis. Since the parasites are likely to multiply in the various tissues and organs, it might be possible, in cases of clinical toxoplasmosis, to find tachyzoites in lymph aspirates, bone marrow, bronchoalveolar lavage fluid, cerebrospinal fluid or cutaneous nodules after fine-needle aspiration, centrifugation, staining and cytological reading. Isolation of the parasite by mouse bioassay is rarely applied and limited to specialized laboratories. Body fluid or a lysate of a tissue sample is inoculated intraperitoneally or subcutaneously into a mouse, followed by serological testing for *T. gondii* specific antibodies 4–6 weeks later. It is a laborious and time-consuming technique that is still used for diagnosis in people with immunosuppression or seroconversion during pregnancy [21]. An alternative method, with the same objective, which was applied here, is the cell culture of the lysate of the tissue sample. It is an ethically more acceptable technique, that has several drawbacks compared to mouse bioassay: it is less sensitive, requires a permanent stock of ready-to-use cell lines, and is prone to fungal or bacterial contamination. The isolation of live, viable and infectious parasites unambiguously demonstrates the involvement of *T. gondii* in clinical manifestations of the disease.

The histological examination of hematoxylin and eosin (H&E)-stained tissue sections confirmed findings of asexual protozoan stages and Periodic acid-Schiff (PAS) staining clearly suggested fungal involvement. *N. caninum* tissue cysts are morphologically different from *T. gondii* tissue cysts with a thick wall, up to 4 µm in old cysts compared to a thin wall for the *T. gondii* [19]; however, no tissue cysts were observed in the present study. Immunohistochemistry (IHC), using a *T. gondii* specific primary polyclonal goat antibody further supported the presumptive diagnosis of a cutaneous toxoplasmosis. IHC showed parasite multiplication at numerous sites of the lesions and adjacent to spore- or yeast-like structures. However, it has to be noted that specificity of immunohistochemistry is largely affected by the quality of the primary antibody used and not all polyclonal antibodies available commercially have been tested for cross-reactions to closely related parasites like *N. caninum.* Thus, IHC findings should be further confirmed by techniques to detect the DNA of the parasite. 

In both ways, cytologically and histologically, a fungal infection was demonstrated in addition to *T. gondii*. PCR amplification and subsequent sequencing suggested that the causative fungus belonged to the genus *Alternaria*. This type of filamentous dematiaceous fungus is commonly found in soil or on plants where it may have a pathogenic role. In humans and animals, *Alternaria* species are associated with subcutaneous infections following accidental inoculation through the skin. In a recent study investigating the incidence of opportunistic fungal infections among dogs diagnosed with immune-mediated diseases and treated with immunosuppressive drugs, 67% of them were diagnosed with phaeohyphomycosis (an infection due to dematiaceous fungi including *Alternaria* species) [23]. The definitive diagnosis requires demonstration of fungal structures (yeasts like elements or filaments) in histopathological preparations, isolation on fungal culture or PCR analysis, similar to our case. 

## 4. Conclusions

Clinical signs of cutaneous toxoplasmosis or neosporosis are not specific. Further laboratory confirmation is required. Even though serology can orientate the diagnosis, PCR is the only way to confirm the differential diagnosis between *N. caninum* and *T. gondii*. Co-infections with *T. gondii* and nonfungal agents have already been described but the present case is the first of a unique combination of *T. gondii* and *Alternaria* sp. This case highlights the potential for the development of opportunistic infections, like toxoplasmosis or fungal infection, following immunosuppressive therapy. Treating against toxoplasmosis and reducing the dosage of prednisolone resulted in resolution of the skin lesions, illustrating the importance of inherent immunity against opportunistic infections. Therefore, we currently recommend that dogs and cats should undergo serologic testing for toxoplasmosis or neosporosis prior to immunosuppressive therapy and undergo regular dermatological evaluations during the immunosuppressive therapy.

## Figures and Tables

**Figure 1 pathogens-12-00114-f001:**
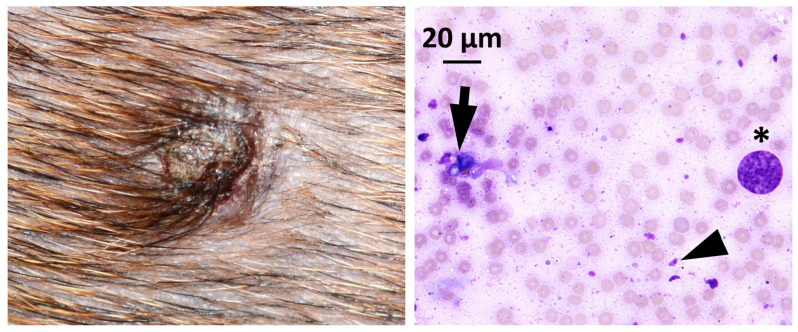
Thorax nodule (left) and fine-needle aspiration cytology (right). Numerous crescent-shape elements (5 × 3 μm) consistent with protozoan tachyzoites (arrow head), yeast-like elements (arrow) and a parasitophorous vacuole filled with tachyzoites (star) were observed.

**Figure 2 pathogens-12-00114-f002:**
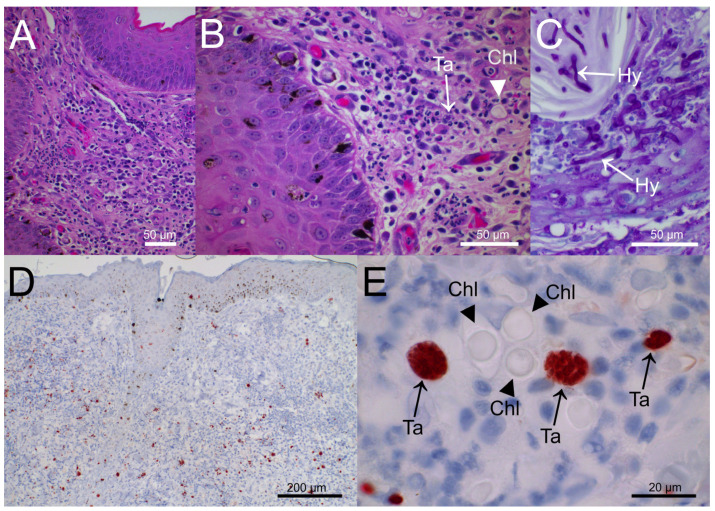
Punch biopsy of the thorax nodule (**A**). Focally extensive pyogranulomatous inflammation of the upper dermis with scattered 10 µm diameter spheric microorganisms and moderately hyperplastic epidermis (HE stain). (**B**). Perivascular to diffuse pyogranulomatous dermatitis; few tachyzoites (Ta, arrow) and spherical structures evocative of fungal chlamydospores (Chl, arrowhead) are visible (HE stain). (**C**). Heavy colonization of hair (endothrix) and hair shaft by PAS-positive fungal filaments (hyphae, Hy, arrows). (**D**). Immunohistochemistry for *Toxoplasma gondii*, numerous immunolabelled (brown) protozoan tachyzoites scattered throughout dermis and epidermis. (**E**). Spheric fungal structures (chlamydospore-like elements, Chl, arrowheads) and immunolabelled (brown) parasitophorous vacuoles filled with tachyzoites (Ta, arrows).

## Data Availability

Not applicable.

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
