# Peer review of "Toxoplasma gondii and Alternaria sp.: An Original Association in an Immunosuppressed Dog with Persistent Skin Lesions"

_pathogens, 2023, doi:10.3390/pathogens12010114_

Round 1

Reviewer 1 Report

In this case report, the coinfection of Toxoplasma and Alternaria is demonstrated in a dog under immunosuppression treatments. While the study itself is pretty straightforward, the research is sound and the case explained clearly. In addition, there is an appropriate background and discussion to integrate the results. Hence, I have no further comments or suggestions and I endorse its publication in Pathogens. Nevertheless, I have one point that I would like to raise and know authors opinion about it. Despite finding Toxoplasma DNA in both the tissue and culture samples, have authors considered performing a Neospora-specific PCR to unequivocally rule out a mixed infection with minor amounts of Neospora that could not have been detected by IHC or in vitro culture? Even better, it would have been great to perform a PCR of a conserved region (e.g. 28S) that can amplify DNA from several apicomplexan parasites and later on sequence the amplified PCR product to see what parasite/s were present. I encourage authors to briefly discuss this issue in their manuscript. 

Reviewer 2 Report

The manuscript is written in a clear and logical manner. The images are of good quality. Attention was given to details as the images were clearly labeled and annotated. 

I only have one critique. The method of PCR-RFLP genotyping was not clearly stated. How many markers did you use? What is your result? You cited the Andreopoulou et al., 2022 publication, however, the original methodology of PCR-RFLP was described in Su et al., 2016 paper. Please correct and clearly state your methodology even if it's from a publication. Please also use the 10 genetic markers to genotype your strain and report the ToxoDB number of the genotype. You should also discuss where this genotype was found and draw upon any previous data/publications/research findings to understand its distribution, virulence, etc.
